# Recombination and Pol ζ Rescue Defective DNA Replication upon Impaired CMG Helicase—Pol ε Interaction

**DOI:** 10.3390/ijms21249484

**Published:** 2020-12-13

**Authors:** Milena Denkiewicz-Kruk, Malgorzata Jedrychowska, Shizuko Endo, Hiroyuki Araki, Piotr Jonczyk, Michal Dmowski, Iwona J. Fijalkowska

**Affiliations:** 1Institute of Biochemistry and Biophysics, Polish Academy of Sciences, Pawinskiego 5a, 02-106 Warsaw, Poland; mdenkiewicz@ibb.waw.pl (M.D.-K.); mmajewska@ibb.waw.pl (M.J.); piotrekj@ibb.waw.pl (P.J.); 2National Institute of Genetics, 1111 Yata, Mishima, Shizuoka 411-8540, Japan; sendo@nig.ac.jp (S.E.); hiaraki@nig.ac.jp (H.A.)

**Keywords:** CMGE helicase-polymerase complex, GINS complex, *PSF1*/*GINS1*, DNA polymerase epsilon, Pol ε, polymerase zeta, Pol ζ, recombination, DRIM, DNA replication fidelity, genetic instability, repeat tracts instability

## Abstract

The CMG complex (Cdc45, Mcm2–7, GINS (Psf1, 2, 3, and Sld5)) is crucial for both DNA replication initiation and fork progression. The CMG helicase interaction with the leading strand DNA polymerase epsilon (Pol ε) is essential for the preferential loading of Pol ε onto the leading strand, the stimulation of the polymerase, and the modulation of helicase activity. Here, we analyze the consequences of impaired interaction between Pol ε and GINS in *Saccharomyces cerevisiae* cells with the *psf1-100* mutation. This significantly affects DNA replication activity measured in vitro, while in vivo, the *psf1-100* mutation reduces replication fidelity by increasing slippage of Pol ε, which manifests as an elevated number of frameshifts. It also increases the occurrence of single-stranded DNA (ssDNA) gaps and the demand for homologous recombination. The *psf1-100* mutant shows elevated recombination rates and synthetic lethality with *rad52Δ*. Additionally, we observe increased participation of DNA polymerase zeta (Pol ζ) in DNA synthesis. We conclude that the impaired interaction between GINS and Pol ε requires enhanced involvement of error-prone Pol ζ, and increased participation of recombination as a rescue mechanism for recovery of impaired replication forks.

## 1. Introduction

Faithful replication of the genome and precise segregation of sister chromatids are essential for the maintenance of chromosome stability. External or endogenous impediments that stall or slow replication forks are sources of replication stress. Replication stress influences the maintenance of genome integrity and is an important source of various physiological pathologies, including cancer [1,2].

Precise DNA replication requires coordinated action of the catalytic and noncatalytic proteins forming the replisome complex [3]. In recent years, in vitro reconstitution studies, as well as structural and single-molecule analyses, have provided much data regarding the formation of complexes involved in helicase loading, activation, and replication fork elongation; for review, see [4,5,6]. In parallel, there is still a need for in vivo data analyzing functional interrelationships and the role of individual replisome subunits in the safeguard of genomic stability. However, such analyses are challenging, due to the difficulty in obtaining viable strains with mutated alleles in genes encoding these essential proteins.

Replication machineries assembled at the replication origin move by the action of 11-subunit CMG helicase, which consists of Cdc45 [7,8,9], the heterohexameric Mcm2–7 helicase motor ring [10,11,12], and the GINS heterotetramer (Psf1, Psf2, Psf3, Sld5) [13,14,15]. Despite being a helicase engine that unwinds DNA to enable synthesis of daughter strands, the Mcm2–7 complex, without cooperation with other subunits, exhibits poor helicase activity [16]. Mcm2–7 activity is stimulated by association with Cdc45 and GINS required for initiation and fork progression. In addition to the important function supporting helicase activity, CMG acts as a platform ensuring numerous interactions of proteins cooperating with the replisome, reviewed in [17]. DNA synthesis is carried out by three multisubunit DNA polymerases, Pol α (alpha), Pol δ (delta), and Pol ε (epsilon), which have distinct roles in the replication fork, reviewed in [18]. DNA Pol α has primase and polymerase activity. Pol δ is involved in lagging strand synthesis [19,20,21]. However, a number of studies show that Pol δ also plays an important role in initiating leading strand synthesis and may take over synthesis on the leading strand when DNA replication by Pol ε is impeded [22,23,24,25,26]. Pol ε is responsible for replication of the leading strand [27] but was additionally postulated to link DNA replication with the S-phase checkpoint in *Saccharomyces cerevisiae* [28,29,30,31,32]. Pol ε and CMG are associated structurally and functionally, forming the CMGE complex, with CMG having a stimulating activity on the polymerase [21,33,34,35,36,37,38]. Moreover, the GINS complex contributes to the preferential loading of Pol ε on the leading strand [21,39]. Additionally, Pol ε modulates the activity of the CMG helicase when the replication fork encounters a barrier [40]. Among the subunits of Pol ε, Pol2 is catalytic and essential [41,42], whereas Dpb2, Dpb3 and Dpb4 are noncatalytic, with only Dpb2 being essential [43,44,45]. Recently, it has been shown that Dpb2 directs the leading strand from the helicase to the active site of Pol ε [46]. Additionally, Dpb2, through its interactions with the Psf1 subunit of GINS, plays the central role in the association between Pol ε and GINS [35,39,47,48,49,50].

Under circumstances disturbing the replication process, DNA synthesis may be continued by specialized polymerases involved in trans-lesion synthesis (TLS). In yeast cells, Pol ζ (zeta), a TLS polymerase, is composed of the Rev3 subunit with a DNA polymerase domain and the accessory subunit Rev7 [51]. The two other accessory subunits, Pol31 and Pol32 of Pol ζ, are shared with Pol δ [52,53,54]. Pol ζ is responsible for the majority of DNA damage-induced mutagenesis, as it is involved in the direct bypass of several types of lesions, reviewed in [55]. Moreover, due to the lower intrinsic fidelity of Pol ζ and its ability to introduce errors while replicating undamaged DNA, reviewed in [56], this polymerase produces 30–70% of spontaneous mutations [49,57,58,59]. Studies have also shown that a significant fraction of defective replisome-induced mutagenesis (DRIM) depends on the *REV3* gene. DRIM occurs as a result of mutations in replicative polymerases or accessory subunits that impair the proper functioning of the replisome [49,60,61,62,63,64,65,66].

Alongside the contribution of error-prone polymerases and various DNA repair mechanisms, homologous recombination (HR) is central to the maintenance of genome stability in all eukaryotic organisms. Through identification of a homologous sequence this mechanism enables the restoration of genetic information and facilitates the repair of DNA breaks or the recovery of stalled replication forks. The highly conserved proteins Rad52 and Rad51 are the major recombinases. Rad52 is the mediator facilitating the recruitment of Rad51 to replication protein A (RPA)-coated single-stranded DNA (ssDNA). This enables the formation of the Rad51/ssDNA nucleofilament involved in searching for and invading the homologous DNA sequence, ssDNA annealing and DNA strand exchange, reviewed in [67,68]. HR is important for both S-phase fork impediment bypass and for the postreplicative ssDNA filling processes, reviewed in [69,70].

The study of noncatalytic proteins of the CMGE complex is attracting greater attention, because mutations in genes encoding its subunits or changes in their expression have a significant impact on the development of cancer or genetic diseases [71,72,73,74,75,76,77,78,79]. In contrast to the role of DNA polymerases, which have been analyzed in detail, the influence of the noncatalytic components of the replisome on the DNA replication process remains poorly understood. Moreover, the numerous proteins of CMGE form complex interactions, and the disruption of either of them may severely affect DNA replication, leading to genomic instability. To characterize the physiological consequences of the impaired interaction within CMGE, we employed the *S. cerevisiae psf1-100* allele constructed in our laboratory [49]. The Psf1-100 subunit possesses substitution of four highly conserved amino acids (V161A, F162A, I163A and D164A) in the C-terminal region, the B domain of Psf1. The *psf1-100* strain exhibits a cold-sensitive phenotype (inability to grow at 18°C) and dumbbell cell morphology with the nucleus in the isthmus between mother and daughter cell. Previously, using two-hybrid analysis and pull-down assays, we showed that the *psf1-100* mutation strongly impairs the interaction of Psf1 with the N-terminal part of the Dpb2 subunit of Pol ε [49]. Therefore, we sought to determine the effects of this impairment on the assembly and functioning of the CMGE complex including possible consequences for the stability of the genome.

## 2. Results

In eukaryotes, the CMG helicase complex and the leading strand polymerase Pol ε form a 15-subunit assembly called CMGE [33,36]. To broaden the knowledge of the mechanism of coupling DNA unwinding with synthesis, we used the Psf1-100 mutant form of the GINS subunit with V161A, F162A, I163A, and D164A substitutions located in the C-terminal region, the B domain. Importantly, it has been shown that the B domain of Psf1 associates with the Dpb2 subunit of Pol ε [39].

### 2.1. Psf1-100 Impairs the Interaction between GINS and Pol ε, Which Results in Impeded DNA Synthesis

To verify the impact of Psf1-100 on DNA replication processes first, we used purified GINS comprising Psf2, Psf3, Sld5, and Psf1 or Psf1-100 subunits, as well as Pol ε comprising Pol2, Dpb2, Dpb3, and Dpb4 subunits, to analyze the interactions between these two complexes (Figure 1A). After incubation, protein mixtures were separated on sodium dodecyl sulfate polyacrylamide gel electrophoresis (SDS-PAGE), followed by immunodetection of Dpb2 and Psf2 subunits. In the control reaction, we detected bands representing the subunits of both GINS and Pol ε (Figure 1A), demonstrating that under the experimental conditions that were used, GINS binds Pol ε. Similar results were observed for the Psf1-1 mutant subunit (R84G) [13]—the band intensity was 0.96 compared to Psf1. In contrast, when GINS contained the Psf1-100 subunit, the band intensity was 0.31, compared to Psf1, while, in absence of Psf1, the band intensity was 0.11 (negative control). This demonstrates impaired interaction between GINS_Psf1-100_ and Pol ε.

To analyze the effect of the *psf1-100* mutation on DNA synthesis, we performed an in vitro replication assay [40,80]. We reconstituted the budding yeast DNA replication machinery using 19 purified replication factors assembled from 52 proteins. Reactions were performed with increasing amounts of GINS_Psf1_ or GINS_Psf1-100_. When wild-type GINS was used, the maximal (100%) replication activity was reached at a 10 nM GINS concentration. At the same GINS concentration, the replication activity of the replisome with Psf1-100 subunit reached only 50% of the maximal activity observed for the native Psf1. The maximal activity of the mutated replisome was achieved only at 60 nM GINS (Figure 1B). This result reflects impaired functioning of the Psf1-100 replisome.

### 2.2. Increased Formation of Single-Stranded DNA in psf1-100 Cells

Perturbations in DNA replication in the *psf1-100* mutant may result in more frequent formation of single-stranded DNA (ssDNA) regions. As ssDNA is coated with RPA [81,82], to evaluate the formation of ssDNA, we analyzed the number of foci formed by Rfa1-YFP, a subunit of RPA. The number of cells with Rfa1 foci was significantly elevated in the *psf1-100* strain compared to the wild-type strain. Moreover, more mutant cells contained multiple Rfa1 foci (Figure 2A and Appendix A). This suggests that in *psf1-100* cells, the formation of ssDNA regions is increased.

### 2.3. Homologous Recombination Rescues psf1-100 Cells

ssDNA regions formed as a result of impaired DNA replication may be repaired by HR mechanisms. Rad52, as a recombination mediator, promotes HR by facilitating the exchange of ssDNA-bound RPA with the Rad51 protein. After strand invasion, Rad51-coated ssDNA searches for sequence homology [67,68]. To verify whether recombination processes are more pronounced in *psf1-100* cells, we analyzed the formation of repair foci by Rad51-GFP and Rad52-YFP. The fusion proteins are recruited to DNA impediments for repair; thus, the frequency of Rad52–YFP or Rad51-GFP foci can be used to monitor the involvement of HR repair [83,84]. We found that the number of spontaneously formed Rad51 and Rad52 foci was higher in the *psf1-100* mutant cells than in the wild-type cells (Figure 2B,C). To investigate whether the frequency of recombination events is changed in *psf1-100* cells, we used a genetic tool elaborated in T. Petes’ laboratory [85]. This system uses a diploid strain, which carries the *can1-100* allele (with an ochre-suppressible mutation) at one *CAN1* locus, and the *SUP4-o* gene (an ochre suppressor) at the second *CAN1* locus on the other chromosome V copy. Recombination events are identified through selection of Can^R^ colonies resulting from the formation of *can1-100* homozygous cells (lacking *SUP4-o*) and *SUP4-o* homozygous cells (lacking the *can1-100* allele). The selection of Can^R^ clones enables the calculation of rates of recombination events. We constructed appropriate *PSF1* and *psf1-100* diploid strains and observed that the rate of wild-type Can^R^ clones formation was 1.51 × 10^−5^, while for *psf1-100* ones, the rate was 10.46 × 10^−5^ (Figure 2D). This result confirms that recombination events are significantly more pronounced in *psf1-100* cells.

Next, we analyzed the genetic interaction between the *psf1-100* mutation and the deletion of the *RAD52* gene encoding the mediator of recombination. Our attempts to construct the double mutant through tetrad dissection from a *psf1-100/PSF1 rad52Δ/RAD52* diploid strain failed and have shown that such a strain is inviable (Figure 2E). Then, we sought to dissect tetrads from a heterozygous *psf1-100/PSF1 rad51Δ/RAD51* diploid strain and obtained haploid double mutant cells. Next, we analyzed the rate of spontaneous mutagenesis in this strain and compared it to single mutants. In this experiment, the Can^R^ mutagenesis rates for *psf1-100* and *rad51Δ* were 220 × 10^−8^ and 498 × 10^−8^, respectively, while in *psf1-100 rad51Δ,* the rate reached 1436 × 10^−8,^ which shows a synergistic effect (Figure 2F). Together, these results points-out the involvement of recombination processes in the maintenance of genomic stability in *psf1-100* mutant cells.

To further characterize Rad52- and Rad51-dependent processes activated in response to replication impediments that arise in *psf1-100* cells, we tested the involvement of the template switch (TS), reviewed in [86], or break-induced replication (BIR), reviewed in [87]. These processes enable completion of DNA replication after replication fork stalling or in the presence of DNA replication impediments [88]. TS uses the newly created sister chromatid as a template for DNA synthesis and requires Ubc13-Mms2-Rad5-dependent polyubiquitylation of the proliferating cell nuclear antigen (PCNA) [86,89,90]. Therefore, we deleted the *MMS2* gene to test whether the TS mechanism contributes to the restoration of DNA synthesis in *psf1-100* cells. The rate of Can^R^ mutagenesis in the double *psf1-100 mms2Δ* showed an additive effect (388 × 10^−8^) compared to the single mutants: 182 × 10^−8^ for *psf1-100* and 282 × 10^−8^ for *mms2Δ* (Figure 2G). This demonstrates that the activity of the TS mechanism is not increased in response to replication dysfunctions in the *psf1-100* mutant.

BIR operates when a single-ended double strand break is created, which after 5′-3′ resection, invades a homologous sequence, reviewed in [87]. This process depends on the activity of the Pif1 helicase and Pol δ [91,92]. Therefore, to investigate the possible involvement of BIR in the response to replication perturbations in *psf1-100* cells, we first analyzed the genetic interaction of this mutation with *PIF1* gene deletion. The double mutant obtained through tetrad dissection demonstrated a 453 × 10^−8^ rate of Can^R^ mutagenesis at the *CAN1* locus (Figure 2G), which represents an additive effect compared to single mutants *psf1-100* (182 × 10^−8^) and *pif1Δ* (328 × 10^−8^). Next, we tried to combine *psf1-100* with deletion of *POL32* encoding a subunit of Pol δ required for BIR [93]. However, our attempts to obtain double *psf1-100 pol32Δ* mutants by tetrad dissection were unsuccessful (Appendix A). To exclude the possibility that this lethal effect results from increased participation of Pol δ in DNA replication in *psf1-100* cells, we used the *pol3-5DV* variant of Pol δ, which is impaired in proof-reading activity [94]. We analyzed the mutagenesis rates in the double mutant *psf1-100 pol3-5DV* in both MMR-deficient and MMR-proficient backgrounds (Appendix A). The non-synergistic effect in an MMR-deficient background may show that the participation of Pol δ in DNA replication in *psf1-100* cells is not increased. Similarly, the same effect in an MMR-proficient background suggests that neither is the involvement of Pol δ in repair synthesis enhanced, at least at the level detectable in this genetic test.

### 2.4. Pol ζ Exerts Its Mutagenic Activity in psf1-100 Cells Mainly in G2 Phase

The mutagenic effect of Pol ζ may be exerted through its activity during DNA replication in the S phase or repair mechanism, which operates mainly in the G2 phase. To verify this, we constructed yeast strains producing Rev3 protein fusion with the Clb2 N-terminal sequences, as previously described [95]. Clb2 is the cyclin that controls the Cdc28/Cdk1 kinase to tightly regulate the cell cycle and is expressed in the G2/M phase and degraded in the G1 phase [96,97,98]. Therefore, expression of the Rev3 fusion proteins under control of the *CLB2* promoter is restricted to the G2 phase. We used these *G2-REV3* gene fusions to replace *REV3* in the *psf1-100* mutant and wild-type cells. To confirm that the N-Clb2-Rev3 fusion protein is produced in G2 phase, we performed Western blot analysis of proteins from *PSF1* and *psf1-100* cells carrying the *CLB2-REV3* fusion. Cells were released from synchrony after α-factor treatment and analyzed throughout the cell cycle. In parallel, we analyzed the progression of the cell cycle by analyzing the DNA content detected by flow cytometry. Using a Clb2-specific antibody, we detected both Clb2 and N-Clb2-Rev3 fusion proteins in G2 phase cells (Figure 3A). This result confirms that the Rev3 protein is specifically produced in G2 phase. Next, similar to Jentsch and coworkers [95], we verified whether the G2-Rev3 fusion is physiologically active and rescues the UV-sensitivity phenotype of *rev3Δ*. Indeed, G2-Rev3 restored the viability of both wild-type and *psf1-100* cells to the level observed in *REV3*-profficient cells (Figure 3B). Finally, since Rev3 is involved in the mutagenesis induced by UV treatment, we tested whether G2-Rev3 complements the defect in UV-induced mutagenesis. We found that UV-induced mutagenesis in both *rev3Δ* and *psf1-100 rev3Δ* cells was restored in the presence of G2-Rev3 and was comparable to the level observed in wild-type and *psf1-100* cells (Figure 3C). Together, these results demonstrate that Pol ζ with G2-Rev3 is functional in the constructed strains.

Then, we analyzed the rate of spontaneous mutagenesis in wild-type and *psf1-100* cells expressing *G2-REV3*. We found that when Pol ζ activity was restricted to the G2 phase, Rev3-dependent mutagenesis was restored in wild-type as well as in *psf1-100* cells (Table 1). This demonstrates that the increased mutagenesis in *psf1-100* cells may result from Pol ζ activity in postreplicative DNA repair when its expression is delayed to G2 phase.

### 2.5. HR and Pol ζ-Dependent Synthesis Play Important Roles in psf1-100 Cells

Our results presented above demonstrate that both Pol ζ-dependent DRIM and Rad51-dependent DNA repair pathways are involved in supporting DNA replication in *psf1-100* cells. Therefore, we decided to verify the effect of combined mutations impairing these mechanisms, i.e., *rev3Δ* and *rad51Δ* in *psf1-100* cells. We constructed a diploid heterozygous *psf1-100/PSF1 rev3Δ/REV3 rad51Δ/RAD51* strain and through tetrad dissection generated haploid triple mutants. Unfortunately, they were poorly viable and our attempts to analyze the mutagenesis rates in the triple mutant were unsuccessful, due to the low counts of viable cells. To investigate the viability of the triple mutant, we prepared yeast cultures in the same way as the mutagenesis level tests and performed more detailed tests. Using PI staining and fluorescence microscopy to identify dead cells [99], we found that the number of living cells observed in all cultures was approximately 50%, a value expected for cells in the stationary phase [100] (Figure 4A green bars). We compared the number of colony-forming units (CFU) with the total cell count. For the *rev3Δ*, *rad51Δ*, or *psf1-100* single-mutant and the wild-type strain, the CFU was 27.2%, 21.4%, 16.2%, and 21.6%, respectively. There was no significant difference between the *psf1-100* mutant and the wild-type strain (Figure 4A blue bars). Deletion of *REV3* in the *psf1-100* strain significantly increased the CFU from 16.2% to 24.4%, while deletion of *RAD51* had no effect, with 12% (Figure 4A blue bars). However, deletion of both *REV3* and *RAD51* in the *psf1-100* strain significantly reduced the CFU count to 5% (Figure 4A blue bars). The CFU value calculated for the *rev3Δ rad51Δ* strain was significantly higher by 15% than for the triple mutant *rev3Δ rad51Δ psf1-100* (Figure 4A blue bars). We also analyzed the DNA content in these strains using flow cytometry with SYTOX Green staining. The *psf1-100* strain demonstrated an increased fraction of cells in the S phase, and similar results were obtained for *psf1-100 rev3Δ* and *psf1-100 rad51Δ* double mutants (Figure 4B). In contrast, deletion of both *REV3* and *RAD51* in *psf1-100* cells resulted in abnormalities in the DNA content, suggesting genomic instability (Figure 4B). These results demonstrate that inactivation of both Pol ζ-dependent synthesis and Rad51-dependent mechanisms has a strong negative effect. Therefore, their functioning is crucial for the survival of *psf1-100* cells.

### 2.6. The psf1-100 Allele Facilitates Primer-Template Rearrangements, Frameshift Formation and the Instability of Repeated DNA Tracts

Impaired interaction of GINS with Pol ε observed in *psf1-100* mutant cells may significantly affect functioning and stability of Pol ε within the replisome. Dissociation of the replicase from the growing point of replication has been proposed to cause misaligned reassociation of the primer with the template leading to polymerase slippage [101]. To obtain detailed information on the specificity of Pol ε errors and possible polymerase slippage enhanced by the mutant form of GINS, we analyzed the spectrum of mutagenesis at the *CAN1* locus, which enables the analysis of a wide range of mutational events. To do this, we combined *psf1-100* with the *pol2-4* allele in the *rev3Δ* background. The Pol2-4 (D290A, E292A) catalytic subunit of Pol ε lacks the proofreading function; thus, Pol ε proofreading does not contribute to error removal in strains carrying this allele [102]. Additional deletion of *REV3* prevents the introduction of errors by Pol ζ which demonstrated increased participation in DNA synthesis in *psf1-100* cells [49]. The results of mutation rate analyzes obtained for *psf1-100 pol2-4 rev3Δ* were compared with spectra of *rev3Δ*, *psf1-100 rev3Δ*, and *pol2-4 rev3Δ* (Figure 5 and Appendix A). We observed a synergistic effect of *pol2-4* and *psf1-100* alleles in the *rev3Δ* background (Appendix A), suggesting that two different mechanisms act in concert to influence the pool of errors. The class of mutations that were highly represented among base substitutions in the *psf1-100 pol2-4 rev3Δ* strain were AT→TA transversions with a mutagenesis rate of 74 × 10^−8^, compared to the ≤0.5 and 25 × 10^−8^ rates obtained for the *psf1-100 rev3Δ* and *pol2-4 rev3Δ* strains, respectively (Figure 5 and Appendix A). These differences were statistically significant with (*p*-values ≤ 0.05). However, the most pronounced increase in mutation rates in *psf1-100 pol2-4 rev3Δ* compared to *psf1-100 rev3Δ* and *pol2-4 rev3Δ* was +1 insertion with mutagenesis rates 86, 2.5, and 13 × 10^−8^, respectively, with *p*-values ≤ 0.05 (Figure 5 and Appendix A). These results indicate increased polymerase slippage in *psf1-100* cells.

DNA polymerase slippage during replication may be the source of instability of repeated tracts. Such DNA regions, also called satellite sequences, consist of up to several hundred repeats of 1 to 100 base pairs [103]. They are frequently found in genomes [104] and have an impact on genetic regulation and chromatin organization. Their instability has been correlated with numerous diseases, e.g., Huntington’s disease, myotonic dystrophy, spinocerebellar ataxia, progressive myoclonus epilepsy, attention-deficit hyperactivity disorder, and cancer [105]. To test whether impaired interaction between GINS and Pol ε may cause instability of repeated tracts, we used the plasmid-based frameshift assay elaborated in T. Petes’ laboratory [106]. The DNA repeat tracts were cloned in frame with the *URA3* gene coding sequence. In this analysis, we used plasmids with repeat tracts as follows: (G)_18_, (GT)_25_, (AACGCAATGCG)_4_, and (CAACGCAATGCGTTGGATCT)_3_ [107,108]. Additionally, as a control, we used a plasmid with a random nucleotide sequence inserted in frame in the *URA3* gene [109]. Expansion or contraction of the repeated sequence results in out-of-frame insertion or deletion and, therefore, incorrect translation of *URA3*. Yeast with such modifications are selected on media containing 5-fluoroorotic acid (5-FOA), which is toxic to cells producing the Ura3 protein [110]. For the control sequence, which reflects the level of forward mutagenesis, 5-FOA-resistance mutations appeared 3.1-fold more frequently in *psf1-100* than in wild-type cells (Figure 6 and Appendix A). A significant increase in repeated tract instability in *psf1-100* was observed for the (G)_18,_ (GT)_25_, (AACGCAATGCG)_4_, and (CAACGCAATGCGTTGGATCT)_3_ sequences with 3.0-, 6.3-, 2.1-, and 4.3-fold changes, respectively (Figure 6 and Appendix A). The obtained results indicate that impaired interaction between CMG and Pol ε significantly increases repeat tract instability. This instability may result from polymerase slippage but other mechanisms, such as HR, cannot be excluded. However, testing the later hypothesis is difficult since the *psf1-100* mutation shows synthetic lethality with *RAD52* deletion.

## 3. Discussion

Smooth replication fork progression depends on coordinated DNA unwinding and synthesis by the replisome components. In addition to in-depth research into the role of DNA polymerases in controlling DNA replication fidelity, several studies indicate the significant role of other noncatalytic components of the replisome in genome stability [49,61,62,64,66,111]. An important role in controlling the correct course of replication and the recruitment of relevant replisome factors, including the leading strand polymerase Pol ε, is played by the CMG helicase complex [7,8,9,10,11,12,13,14,15]. It was shown that the Psf1 subunit of GINS, an essential structural component of CMG helicase, interacts with Dpb2, an essential noncatalytic subunit of Pol ε [39,47,48,49,50,112]. The interaction between CMG and Pol ε is critical not only for appropriate assembly of the initiating and elongating complex and targeting Pol ε to the leading strand, but also for proper functioning of both.

To examine how important this interaction is for replisome activity and maintaining genetic stability, we employed the *psf1-100* allele isolated in our laboratory [49]. Psf1-100 contains mutations in four highly conserved amino acids in the C-terminal region, the B domain of the protein [49], which was shown to be responsible for interaction with the N-terminal region of Dpb2, an essential Pol ε subunit [39,113]. Here, employing in vitro methods, we demonstrate that impaired interaction between Psf1-100 and Dpb2 severely affects the interaction between GINS_Psf1-100_ and Pol ε complexes (Figure 1A).

Cells with the *psf1-100* allele accumulate in S phase (Figure 4B). They exhibit a cold-sensitive phenotype, and even at the permissive temperature (30 °C), cells form larger, dumbbell cells [49]. The in vitro DNA replication assay performed in this work with GINS_Psf1-100_ shows reduced activity when compared to the wild-type complex (Figure 1B), which suggests impaired DNA replication elongation. Since both GINS and Pol ε are recruited to the pre-replication complex as components of the pre-loading complex [114,115,116,117], we cannot exclude that impaired interactions between GINS and Pol ε in *psf1-100* cells also affect the initiation steps.

The *psf1-100* allele increases the participation of Pol ζ in DNA synthesis, so this mutation can be classified as causing defective replisome-induced mutagenesis (DRIM, see Introduction). Previously, we showed, that the observed Pol ζ-dependent errors in *psf1-100* cells were not corrected by the MMR mechanism [49], suggesting that they appeared outside of the S phase. Moreover, the observed mutator effect of Pol ζ is maintained when its activity is restricted to the G2 phase (Table 1). These two phenotypes support the hypothesis that Pol ζ is involved in repair mechanism-associated DNA synthesis in *psf1-100* cells. Moreover, our results show that in addition to enhancing participation of Pol ζ, the *psf1-100* mutation also affects the functioning of Pol ε. By combining *psf1-100* with the proofreading-deficient *pol2-4* allele in the Pol ζ-deficient background, we confirmed slippage of Pol ε during DNA replication, which manifests by an increased number of + 1 frameshift mutations in the double mutant cells (Figure 5 and Appendix A). Recently, Yuan and coworkers (Yuan 2020) suggested a role of Dpb2 in positioning the leading strand in the replisome. Therefore, it cannot be excluded that impaired interaction between Dpb2 of Pol ε and Psf1-100 of GINS affects the directing of the leading strand from the helicase to the polymerase.

It was shown that some DNA sequences, especially repeated motifs such as micro- and minisatellites, are subject to rearrangements, especially upon fork slowing or stalling caused by “roadblocks” or defective functioning of the replisome including polymerase slippage, and the resulting increased frequency of HR, reviewed in [118]. Indeed, we show that impaired interaction between GINS and Pol ε significantly enhances repeat tract instability (Figure 6 and Appendix A). This effect can be caused by increased polymerase instability leading to its slippage, but we cannot exclude other mechanisms e.g., HR. Moreover, increased formation of Rfa1 foci (Figure 2A) indicates the presence of RPA-bound ssDNA regions, which also implies problems with DNA replication [119]. Together, these results demonstrate a disturbance of the replication process in *psf1-100* cells, with a possible uncoupling of Pol ε and CMG helicase.

Resolving replication problems can be associated with a number of processes, e.g., polymerase and/or template switching, downstream repriming, and arrival of an opposite fork or recombination. Recombination is frequently coupled with a progressing replication fork allowing its repairs and restart. Yeast Rad51 and Rad52 recombinases were detected in both unperturbed and stressed forks. It was demonstrated that they are not specifically recruited to the stalled fork, but they assist the replisome, ready for response through different mechanisms to problems with continuation of DNA replication [120]. The recombinases have a role in the stabilization of stalled replication forks, and in gatekeeper mechanisms important in preventing excessive fork remodeling [121]. Here, we demonstrate that Rad52 is essential for *psf1-100* cell survival (Figure 2E), while the lack of functional Rad51 has a less deleterious effect. The *psf1-100 rad51Δ* mutant survives at the cost of increased levels of mutagenesis, probably caused by the enhanced contribution of Pol ζ to DNA synthesis (Figure 2F). Importantly, simultaneous inactivation of Pol ζ activity and impairment of recombination processes in the triple *psf1-100 rad51Δ rev3Δ* mutant results in low CFU counts and severe DNA content abnormalities (Figure 4A,B). In addition, an increased number of Rfa1 (RPA), Rad52, and Rad51 foci observed in *psf1-100* cells compared to wild-type cells (Figure 2A–C) indicate the critical role of recombination when the interaction between GINS and Pol ε is impaired. Finally, the analysis of recombination events confirms that these processes are about 7-fold more frequent in *psf1-100*, compared to wild-type cells (Figure 2D).

Rad51 and Rad52 participate in the recovery of collapsed replication forks by several pathways. Among them, BIR and TS have been implicated in the restart of perturbed replication forks [87,89,90]. It was shown that the Mcm2–7 complex of replicative helicase does not participate in BIR [122]; instead, another helicase, Pif1, is essential for this process [92]. Here, we demonstrate that neither BIR, nor TS is enhanced in the *psf1-100* strains, as deletion of *PIF1* or *MMS2* (encoding one of the components of the Ubc13-Mms2-Rad5 complex involved in TS [86,89,90,123]) has an additive effect on the rate of mutagenesis in the double mutant (Figure 2G).

Although both recombination and Pol ζ allow continuation of DNA synthesis in the *psf1-100* strain, inactivation of the latter increases the survival of the *psf1-100* strain by approximately 40%, as observed for CFU counts in Figure 4A, and reduces the observed level of mutagenesis to the level of the *rev3Δ* strain (Table 1). To explain these observations we can speculate that, besides the error-prone nature of Pol ζ, DNA repair processes involving this polymerase, due to its lower processivity, can be less extensive than those based on recombination. Interestingly, deletion of *REV3* in the *psf1-100 rad51Δ* strain causes a drastic decrease in CFU counts (Figure 4A). This demonstrates that both switching to Pol ζ and recombination are mechanisms that are employed to fulfill genetic material duplication in *psf1-100* cells.

Therefore, we propose a model (Figure 7) in which impaired interaction of GINS with Pol ε is the source of severe consequences. One of them is polymerase slippage, which results in reduced fidelity of replication proceeded by Pol ε. Moreover, uncoupling of the helicase and polymerase complexes promotes the appearance of ssDNA regions requiring recombination as an essential mechanism enabling survival, promoting replication fork protection by restart and continuation of DNA synthesis. An alternative, very important mechanism ensuring the continuity of genetic material is increased participation of error-prone Pol ζ, which facilitates efficient postreplicative ssDNA gap filling. However, its action reduces the viability of *psf1-100* cells.

Mutations or deregulation of the human homolog of the *PSF1* gene was observed in correlation with disorders involving cancer, neurodevelopmental, and immunodeficiency issues [71,72,73,74,75,76,77]. However, the precise role of GINS subunits in these processes is poorly understood. Therefore, our findings provide important knowledge on the involvement of the GINS complex in the maintenance of genome stability. Importantly, components of the CMG complex were also proposed as targets in innovative anticancer therapies [78,79].

## 4. Materials and Methods

### 4.1. Strains, Media and General Methods

The *S. cerevisiae* strains used in this study (Appendix A) are derivatives of strain ΔI(-2)I-7B-YUNI300 [124]. Yeast were grown at 30 °C in standard media [125]. YPD medium (1% Bacto-yeast extract, 2% Bacto-peptone, 2% glucose liquid or solidified with 2% Bacto-agar) was used when nutrition selection was not required. YPD with appropriate antibiotics (hygromycin B 300 μg/mL (Bioshop, Burlington, ON, Canada), nourseothricin 100 μg/mL (Werner BioAgents, Jena, Germany) or G418 sulfate (geneticin) 350 μg/mL (US Biological, Salem, MA, USA) and SD medium (0.67% yeast nitrogen base without amino acids, 2% glucose, liquid or solidified with 2% Bacto-agar) supplemented with appropriate amino acids and nitrogenous bases were used for transformant selection and mutagenesis assays. SD medium supplemented with 60 μg/mL or 120 μg/mL L-canavanine (Sigma Aldrich, St. Louis, MO, USA) was used to determine the frequency of forward mutations at the *CAN1* locus or frequency of recombination, respectively. SD medium with 1 mg/mL 5-fluoroorotic acid (5-FOA) (US Biological, Salem, MA, USA) was used for the selection of *URA3* mutants [110]. Yeast strains were transformed using the lithium acetate/single-stranded carrier DNA/PEG method [126]. Isolation of chromosomal DNA from yeast was performed using the Genomic Mini AX Yeast Spin Kit (A&A Biotechnology, Gdansk, Poland).

Escherichia coli DH5α (endA1 glnV44 thi-1 recA1 relA1 gyrA96 deoR nupG Φ80d lacZΔ M15Δ (lacZYA-argF) U_169_, hsdR_17_ (r _K_^−^,m _K_^+^), λ- (Invitrogen, California, United States) were grown at 37 °C in LB medium, supplemented when needed with ampicillin 100 μg/mL (PolfaTarchomin S.A., Warsaw, Poland). E. coli cells were transformed as described in [127]. Bacterial plasmids were isolated using the Plasmid Mini Kit (A&A Biotechnology, Gdansk, Poland).

### 4.2. Construction of Yeast Strains

The (*PSF1, LEU2*) and (*psf1-100, LEU2*) cassettes described in [49] were integrated into the *PSF1* locus of the SC765 strain. The presence of the (*PSF1, LEU2*) and (*psf1-100, LEU2*) alleles was confirmed by PCR using primers Inprom and dwPSF1 (Appendix A) and afterward through DNA sequencing. Additionally, the presence of the *psf1-100* allele was verified by a temperature sensitivity test as the *psf1-100* strain does not grow at 18 °C.

Strains carrying deletions of the *REV3, RAD51, RAD52, MMS2, PIF1*, or *POL32* genes were constructed based on the Y1000 or Y1012 strains as previously described in [109] using the primers listed in Appendix A. Strains with the *psf1-100* allele and single or double deletions were constructed by tetrad dissection from diploid strains constructed by crossing strains SC778, SC803, or Y1006 with appropriate single gene deletion *MATα* strains listed in Appendix A. Gene disruption was confirmed by PCR using the primers listed in Appendix A. The presence of *PSF1* or the *psf1-100* allele was verified as described above.

The (*RFA1-YFP, LEU2*) cassette described in [109] was integrated into the *RFA1* locus of the SC766 and SC778 strains. The presence of the (*RFA1-YFP, LEU2*) allele was confirmed by PCR, using the primers RFA6231R, RFA7367F and YFP9451R (Appendix A).

The SC776 and SC778 strains were transformed with the (*G2-REV3, natNT2*) integration cassette described in [95]. The (*G2-REV3, natNT2*) cassette was PCR-amplified with the primers S1_REV3 and S4_REV3 (Appendix A) using pGIK43 [95] as a template. The presence of *G2-REV3* gene fusions was verified by sequencing the PCR-amplified fragment (primers: Rev3A, Rev3-R3, Rev3up, Rev3_R1, prCLB2; Appendix A).

### 4.3. Protein Purification

Proteins were purified as described previously [40].

### 4.4. Pol ε-GINS In Vitro Interaction Assay

Halo-Sld5 GINS (2 pmol) was incubated with rotation with 10 µL of HaloLink Resin in 1× buffer (25 mM HEPES-KOH pH = 7.6, 200 mM K-acetate, 2 mM Mg-acetate, 10% glycerol, 0.1% Tween 20, 0.01% NP-40) at 4 °C for 30 min. Next, casein (0.8 mg/mL) and BSA (20 mg/mL) were added, followed by further incubation with rotation at 4 °C for 30 min. Then, the beads were washed two times with 400 µL of 1× buffer, mixed with 2 pmol of Pol ε and 2 mg/mL of BSA in 400 µL of 1× buffer and incubated with rotation at 30 °C for 30 min. Afterwards, the beads were washed two times with 400 µL of 1× buffer and mixed with 50 µL of 1× sample buffer (62.5 mM Tris-HCl pH = 6.8, 1.0% SDS, 10% glycerol) and incubated at 65 °C for 10 min. The supernatant was mixed with DTT and loading dye and subjected to sodium dodecyl sulfate polyacrylamide gel electrophoresis (SDS-PAGE). Separated proteins were transferred to membranes and incubated with anti-Psf2 [48] and anti-Dpb2 [128] antibodies detected, using an Odyssey infrared imaging system (LI-COR).

### 4.5. In Vitro Replication Assay

pARS1 (245 bp ARS1 fragment cloned into the SmaI site of pNEB193) DNA was mixed with the loading complex composed of 11 nM ORC, 23 nM Cdc6, and 50 nM MCM–Cdt1 in a buffer containing 25 mM HEPES-KOH at pH = 7.6, 100 mM Kglutamate, 10 mM magnesium acetate, 0.01% NP-40, 100 μg/mL of BSA, 1 mM DTT, and 5 mM ATP. After 20 min of incubation at 30 °C, DDK was added directly to the reaction to a final concentration of 26 nM, and incubation was continued at 30 °C for 20 min. Next, the replication complex proteins and solutions: were added directly to the reaction: 25 nM Sld3–Sld7, 30 nM Sld2, 30 nM Dpb11, 30 nM GINS, 40 nM Cdc45, 20 nM Polε, 5 nM Mcm10, 100 nM RPA, 3.4 nM CDK, 5 nM Polα, 20 nM RFC, 20 nM PCNA, 10 nM Top2, 10 nM Polδ, 20 mM each NTP, 8 mM dATP, dCTP, and dGTP, 6 mM dTTP, and 1 mM Biotin-16-dUTP (Sigma-Aldrich). After incubation at 30 °C for 20 min, the reaction was terminated by the addition of 1/5 volume of Alkaline stop dye (loading buffer) containing 0.3 N NaOH, 6 mM EDTA, 36% glycerol, and 0.1% Orange G dye. The products were separated on 1% alkaline agarose gels in 0.05 N NaOH and 1 mM EDTA for 75 min at 75 V. DNA was transferred from the gels to Hybond N+ membranes (GE) in 0.5 TBE Buffet for 30 min at 80 V. The membrane was treated with 20× SCC and crosslinked in UV linker. After treatment with 2% ECL Advance Blocking Reagent in TBST for 10 min and two 5 min washes by TBST, the membranes was incubated for 30 min with 0.5 μg/mL of IR-Dye 680 RD streptavidin (LICOR) in TBST with 10% SDS. After washing in TBST with 0.1% SDS, the membrane was scanned on an Odyssey infrared imaging system (LI-COR).

### 4.6. Measurement of Spontaneous Mutation Frequency at the CAN1 Locus

To determine spontaneous mutation frequencies, 10–20 cultures of 2 or 3 independent isolates of each strain were inoculated in 2–20 mL of liquid SD medium, supplemented with the required amino acids and nucleotides. Cultures were grown at 30 °C to the stationary phase. Aliquots of concentrated cultures and appropriate dilutions were plated on selective (containing L-canavanine) and nonselective media, respectively. After 3–5 days of growth at 30 °C, colonies were counted. The frequency of forward mutations at the *CAN1* locus were calculated by dividing the cell count from selective media by the cell count from nonselective media. Each experiment was repeated at least three times. The spontaneous mutation rates were determined as described below.

### 4.7. Calculation of Mutation Rates and Statistical Analysis

The mutation rates were calculated using the equation *μ = ƒ/ln(Nμ)*, where *μ* is the mutation rate per round of DNA replication, *ƒ* is the mutant frequency, and *N* is the total population size [129]. To calculate the median values of the mutation rates and 95% confidence intervals, STATISTICA 6.0 was used. To determine the *p*-values of the differences between the mutation rates of the respective strains, a nonparametric Mann–Whitney U test was used.

### 4.8. CAN^R^ Mutation Spectrum

A total of 192 cultures of 2 independent isolates of the SC660 strain were inoculated in 1 mL of liquid SD medium supplemented with the required amino acids and nitrogenous bases, lacking uracil and leucine, and grown at 30 °C. When cultures reached the stationary phase, appropriate dilutions were plated on solid medium supplemented with L-canavanine. After 5 days of incubation at 30 °C, total DNA from 192 single CAN^R^ colonies selected randomly from each plate was isolated and used for PCR amplification of the *CAN1* locus with primers MGCANFF and MGCANRR and DNA sequencing with primers Can_1666, Can_1963, Can_2241, and Can_2465 (Appendix A). Changes within the *CAN1* gene were identified using Clone Manager 9 software. For statistical analysis, a contingency table and the χ^2^ test were used.

### 4.9. Detection of Recombination Events

Recombination events were calculated using a previously described method [85]. Single colonies of diploid strains Y1035 and Y1036 were grown on YPG medium at 30 °C for 3 or 4 days, respectively. Next, independent colonies were picked, resuspended in water, and plated on SD medium lacking arginine and supplemented or not with L-Canavanine (120 μg/mL). After 4–6 days of growth at 30 °C, colonies were counted. The frequency of recombination was calculated by dividing the cell count from selective media by the cell count from nonselective media. Each experiment was repeated at least three times. The spontaneous mutation rates were determined, as described previously [85,130].

### 4.10. Determination of UV Radiation Sensitivity and UV-Induced Mutagenesis at the CAN1 Locus

To determine the UV sensitivity of yeast strains, 10–12 cultures of 2 or 3 independent isolates of each strain were inoculated in 20 mL of liquid SD medium (supplemented with required amino acids and nucleotides) and grown at 30 °C to the logarithmic phase. Aliquots of concentrated cultures and appropriate dilutions were plated (in duplicate) on selective (containing L-canavanine) and nonselective media, respectively, and immediately exposed to specified UV doses of 0, 5 and 15 J/m^2^ using a UV crosslinker (UVP model CL-1000). After 3–4 days of growth at 30 °C in the darkness, the colonies were counted. The mutant frequency was calculated by dividing the Can^R^ mutant count by the viable cell number. The results represent the median values. The experiment was repeated three times.

### 4.11. Synchronization in G1 Phase with α-Factor

Strains Y1023 and Y1024 were grown until the OD_600_ reached 0.4. Next, cells were harvested and resuspended in fresh medium supplemented with α-factor (4 µg/mL). Additionally, α-factor (4 µg/mL) was added after 60–90 min of incubation. After 2–3 h of growth at 30 °C, to release them from α-factor, cells were harvested and washed three times with sterile water, and then resuspended in fresh SD medium and incubated at 30 °C; 1 mL samples were immediately collected and fixed in 70% ethanol. Other samples were taken at the indicated time points and fixed in 70% ethanol.

### 4.12. Flow Cytometry Analysis

Samples were prepared for flow cytometry, as described previously [29], with modifications according to specific strain requirements. Yeast cells were stained using propidium iodide (PI) (16 μg/mL) (Sigma Aldrich, St. Louis, MO, USA) (strains Y1023, Y1024) or SYTOX Green (0.5 µM) (Invitrogen, Carlsbad, CA, USA) (strains Y1006, Y1012, Y1013, Y1014, Y1017, Y1018, Y1019, Y1020). The DNA content was determined by measuring the PI or SYTOX Green fluorescence signal (FL2 or FL1, respectively) using Becton Dickinson FACS Calibur and CellQuest software (BD Bioscience, San Jose, CA, United States).

### 4.13. Western Blot Analysis

The Y1023 and Y1024 strain cells were synchronized with the α-factor (see Synchronization in G1 phase with α-factor) and released from G1-arrest into fresh SD medium and incubated at 30 °C. Samples were taken at the indicated time points and prepared, as described previously [131]. Protein extracts were separated by SDS-PAGE (10% polyacrylamide gel) and then transferred onto polyvinylidene fluoride (PVDF) membranes (Millipore, Bedford, MA, USA) in a wet blot system. The blotting was performed in transfer buffer at a constant voltage of 30 V for 16 h at 4 °C. The membrane was blocked with 3% milk in TBST for at least 30 min. Next, the membrane was washed 2 times for 10 min with TBST and incubated for 2 h with primary antibodies diluted in TBST with 1.3% milk. Afterwards, the membrane was washed 2 times for 10 min with TBST and incubated for 1 h with secondary antibodies diluted in TBST with 1.3% milk. Finally, the membrane was washed 2 times for 10 min with TBST. The signal was detected using chemiluminescence of the substrate for horseradish peroxidase (HRP), SuperSignal West Femto Maximum Sensitivity Substrate (Thermo Scientific, Waltham, MA, USA), and documented with a charge-coupled device camera (FluorChem Q Multi Image III, Alpha Innotech, San Leandro, CA, USA). Images were analyzed using ImageJ software (NIH, USA). The N-terminal 180 amino acid fragment of Clb2 was detected with rabbit polyclonal anti-Clb2 (1:5000, y-180, Santa Cruz Biotechnology, Dallas, TX, USA) and goat anti-rabbit IgG conjugated to HRP (1:20,000, sc-2004, Santa Cruz Biotechnology, Dallas, TX, USA). Actin was detected using a mouse anti-actin monoclonal antibody (1:10,000, MAB1501, Millipore, Bedford, MA, USA) and a goat anti-mouse IgG conjugated to an HRP (1:20,000, P0447, DAKO, Santa Clara, CA, USA) antibody.

### 4.14. Identification of Rad52, Rad51 and Rfa1 Foci by Fluorescence Microscopy

Strains SC766 and SC778 were transformed with the pWJ1344 plasmid carrying a *RAD52–YFP* fusion (*LEU2, ARS-CEN, RAD52-YFP, Amp^R^, oriC*) [132] or with pSFP119 plasmid (kindly provided by Steve Jackson) carrying a *RAD51-GFP* fusion (*TRP1*, *RAD51* with 1000 bp upstream promoter sequence and C-terminal GFP-tag cloned into ApaI-XhoI digested pRS414). The *RFA1–YFP* fusion was introduced into the native *RFA1* locus by gene replacement. Cells were grown at 30 °C to the logarithmic phase in SD liquid medium supplemented with the required amino acids and nitrogenous bases lacking leucine or tryptophan. The Rad52, Rad51, and Rfa1 foci were examined using the Axio Imager M2 fluorescence microscope with the AxioCam MRc5 Digital Camera (Zeiss, Oberkochen, Germany). Images were analyzed with Axio Vision 4.8 software. The number of cells and Rad52, Rad51, or Rfa1 foci in the cells were counted. To analyze the possible differences between wild-type and *psf1-100* strains in foci formation, we used contingency tables, and further applied the χ^2^ test. For each strain, three biological replicates were analyzed.

### 4.15. Yeast Viability Assessment

To determine the viability of the strains Y1012, Y1013, Y1014, Y1017, Y1018, Y1019, Y1020, and Y1006, 3 mL cultures of independent isolates of each strain in liquid SD medium supplemented with the required amino acids and nucleotides were grown at 30 °C to the stationary phase. Yeast viability assessment was performed using two methods. First, to calculate the colony-forming units (CFU), cells were counted in a Neubauer counting chamber, and next, the same number of cells (250 cells) was plated in triplicate on nonselective media. After 4 days of growth at 30 °C, the colonies were counted. The CFU was calculated by dividing the number of colonies formed after plating by the total cell number. Second, to calculate viable cells, 1 mL of each culture was harvested by centrifugation, resuspended in 1 mL PBS containing PI (0.32 μg/mL) (Sigma Aldrich, St. Louis, MO, USA) and incubated for 15 min at room temperature in the dark. Next, the samples were again centrifuged, and the pellets were resuspended in 30 μL PBS. Dead PI-stained cells were examined using the Axio Imager M2 fluorescence microscope (Zeiss, Oberkochen, Germany). Images were analyzed with Axio Vision 4.8. At least 800 cells were examined to count the number of dead and viable cells of each strain, and the average percentage of viable cells was calculated.

### 4.16. Stability of Repetitive Sequences

The stability of repetitive sequences was determined, as described previously [109], with modifications. Plasmids pMD28 (18 × 1 nt), p51GT (25 × 2 nt), pMD41 (4 × 11 nt), pEAS20 (3 × 20 nt) [107,108] and pKK2 [109] were used for the transformation of yeast strains SC801 and SC803. To determine mutation rates, 10–20 cultures of 2 independent isolates of each strain were inoculated in 2 mL of liquid SD medium, supplemented with the required amino acids and nitrogenous bases, lacking tryptophan and leucine, and grown at 30 °C. When cultures reached the stationary phase, appropriate dilutions were plated on selective (containing 5-FOA for selection of *URA3* mutants) and nonselective media. Colonies were counted after 3–5 days of incubation at 30 °C. The spontaneous mutation rates were determined as described above.

## Figures and Tables

**Figure 1 ijms-21-09484-f001:**
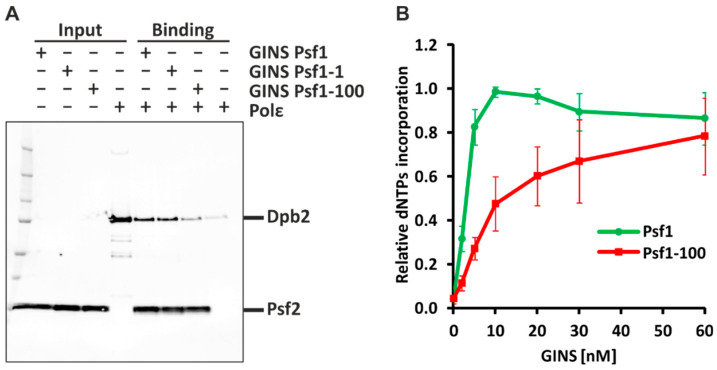
Psf1, Psf2, Psf3, Sld5 complex (GINS)-Pol ε in vitro interaction and DNA replication activity in the presence of GINS_Psf1-100_. (**A**) Western-blot detection of Dpb2 and Psf2 subunits of Pol ε and GINS in protein mixtures separated by sodium dodecyl sulfate polyacrylamide gel electrophoresis (SDS-PAGE). Two pmol of various GINS complexes (WT, Psf1-1 and Psf1-100) containing Halotag-Sld5 were fixed on 10 µL of HaloLink Resin and then incubated with 2 pmol of Pol ε in 400 µL buffer at 30 °C for 30 min. Input of GINS and Pol ε were 0.5 and 0.05 pmol, respectively. Precision Plus Protein™ All Blue Prestained Protein Standard was used as molecular weight marker. Gel patterns of GINS and Pol ε proteins used are shown in Appendix A. (**B**) Reactions performed with increasing amounts of GINS_Psf1_ or GINS_Psf1-100_ were separated on an alkaline agarose gel, and transferred onto a membrane, before the detection of incorporated biotin-UTP using IR dye-labeled streptavidin. The signal intensity reflects the amount of incorporated dNTPs (synthesized DNA)—see also Appendix A. The graph shows the relative intensity normalized to the highest value obtained for GINS_Psf1_.

**Figure 2 ijms-21-09484-f002:**
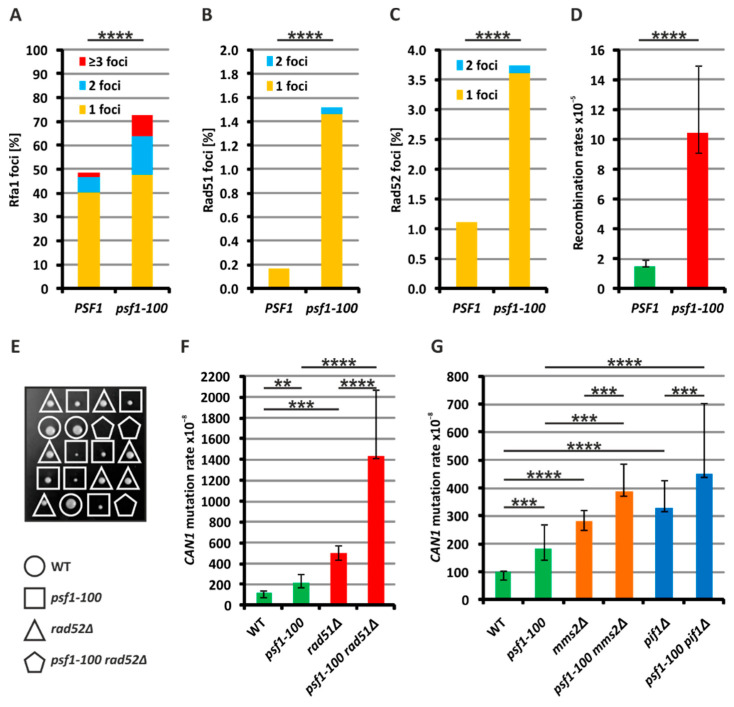
Analysis of single-stranded DNA (ssDNA) formation and homologous recombination mechanisms involved in the response to replication perturbations in *psf1-100* cells. (**A**) Analysis of Rfa1 foci formation (representative images are shown in Appendix A). For each strain, at least 1200 cells from three biological replicates were analyzed. The results represent the number of cells with the indicated number of foci. For statistical analysis, a contingency table and the χ^2^ test were used (Appendix A). The χ^2^ statistic is 236.5692, which corresponds to the *p*-value ≤ 0.0001 (****). (**B**) Analysis of Rad51 foci formation. For each strain, at least 3000 cells from three biological replicates were analyzed. The results represent the number of cells with the indicated number of foci. For statistical analysis, a contingency table and the χ^2^ test were used (Appendix A). The χ^2^ statistic is 56.6504, which corresponds to the *p*-value ≤ 0.0001 (****). (**C**) Analysis of Rad52 foci formation. For each strain, at least 3000 cells from three biological replicates were analyzed. The results represent the number of cells with the indicated number of foci. For statistical analysis, a contingency table and the χ^2^ test were used (Appendix A). The χ^2^ statistic is 49.5449, which corresponds to the *p*-value ≤ 0.0001 (****). (**D**) Rates of recombination events in wild-type and *psf1-100* cells. Medians with 95% confidence intervals were calculated from at least 20 independent colonies. The Mann–Whitney U test was used to determine the *p*-value ≤ 0.0001 (****). (**E**) Synthetic lethality of the *psf1-100* mutation and *RAD52* deletion. Dissection of tetrads from the *psf1-100/PSF1 rad52Δ/RAD52* strain. (**F**,**G**) Spontaneous mutation rates measured in the *psf1-100 rad51Δ* strain (F), *psf1-100 mms2Δ* and *psf1-100 pif1Δ* strains (G). The presented values are medians with 95% confidence intervals calculated from at least ten independent cultures. The Mann–Whitney U test was used to determine the *p*-value ≤ 0.0001 (****); ≤0.001 (***); ≤0.01 (**). The *p*-values are shown in Appendix A.

**Figure 3 ijms-21-09484-f003:**
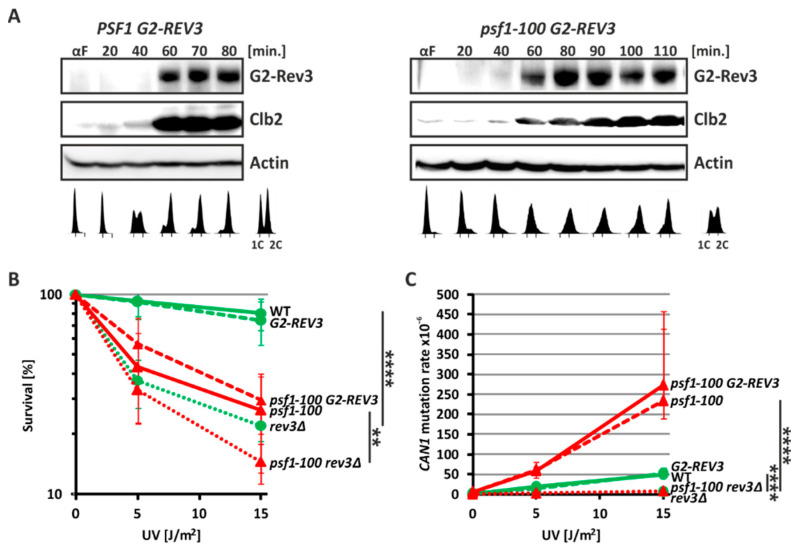
Verification that G2-Rev3 in the constructed strains rescues the phenotypes observed in *rev3Δ* cells. (**A**) Western blot analysis of G2-Rev3 expression. Yeast cells were synchronized with α-factor and released into a new cell cycle. Samples were collected at the indicated times and analyzed using anti-Clb2 and anti-actin antibodies. In parallel propidium iodide staining, FACS analyses of DNA content were performed. (**B**) Sensitivity of wild-type and *psf1-100* cells with *G2-REV3* to UV light at the indicated doses. The mean survival percentage with SD for exponentially growing cells treated with the indicated doses of UV light is shown. For statistical analyses, a T-test was used to determine the *p*-value ≤ 0.0001 (****); ≤0.01 (**), see Appendix A. (**C**) Analysis of UV-induced mutagenesis in wild-type and *psf1-100* cells expressing G2-Rev3. Median values of mutation frequencies with 95% confidence intervals were calculated for exponentially growing cells treated with the indicated doses of UV light. For statistical analyses, a Mann-Whitney U test was used to determine the *p*-value ≤ 0.0001 (****), see also Appendix A.

**Figure 4 ijms-21-09484-f004:**
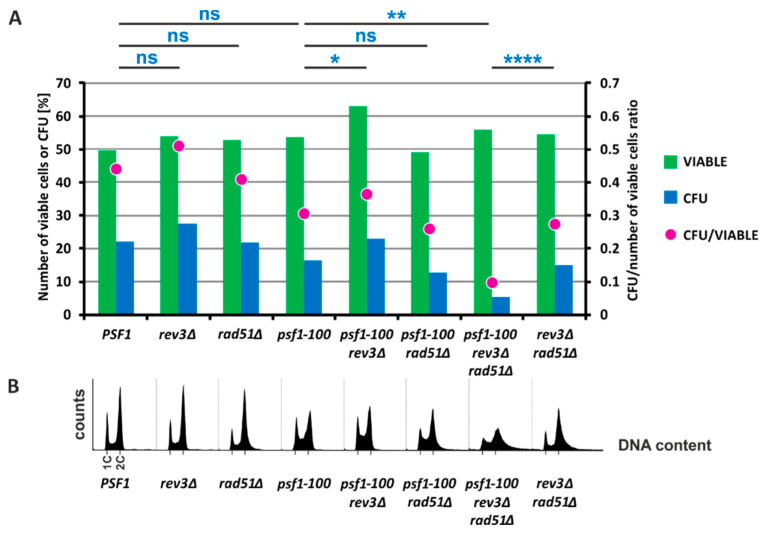
The Pol ζ-dependent error-prone pathway and the homologous recombination-dependent error-free pathway are necessary for the survival of *psf1-100* cells. (**A**) The number of viable cells (green bars) was estimated in stationary phase cultures of the indicated strains by counting propidium iodide (PI)-stained dead cells. In parallel, the number of colonies formed after plating was calculated and compared with the total number of cells (blue bars). A statistical T-test was used to determine the *p*-value ≤ 0.0001 (****); ≤0.01 (**); ≤0.05 (*). The statistical analysis is shown in Appendix A. The proportion of colonies formed from living (PI-negative) cells is shown as purple dots. (**B**) DNA content in asynchronous cells was determined using flow cytometry with SYTOX Green staining. The 1C and 2C DNA content is indicated.

**Figure 5 ijms-21-09484-f005:**
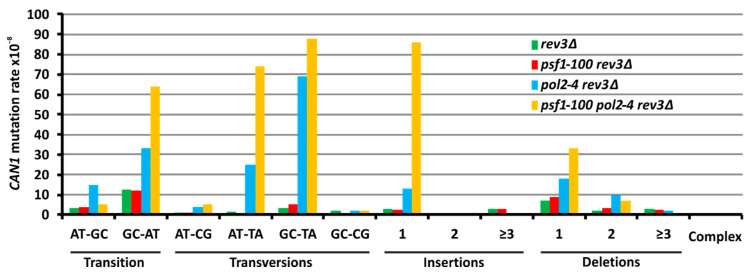
Spectrum of *CAN1* forward mutations for the *psf1-100 pol2-4 rev3Δ* strain. Rates for particular types of base substitutions, insertions, deletions and complex mutations are shown. Data for isogenic *rev3Δ*, *psf1-100 rev3Δ*, and *pol2-4 rev3Δ* strains were already shown previously [49]. Rates, percentages, and relative rates are shown in Appendix A. Statistical analysis results are shown in Appendix A.

**Figure 6 ijms-21-09484-f006:**
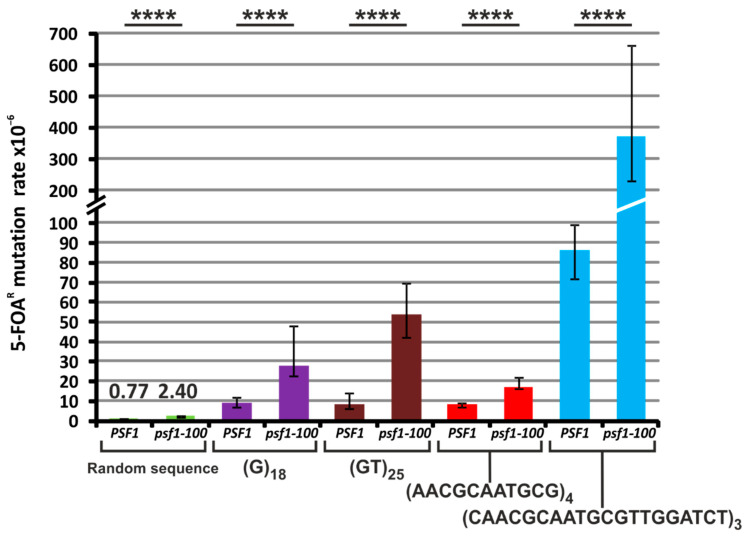
The *psf1-100* allele enhances the instability of various repeat tracts. The 5-fluoroorotic acid (5-FOA)-resistance mutation rates were determined for wild-type and *psf1-100* cells carrying plasmids with the indicated repetitive sequences. Median values with 95% confidence intervals were calculated from data obtained for at least twenty cultures of each strain. The *p*-values for *psf1-100* versus the respective wild-type strains, calculated using the nonparametric Mann-Whitney U statistical test, were ≤0.0001 (****). All associated data are shown in Appendix A.

**Figure 7 ijms-21-09484-f007:**
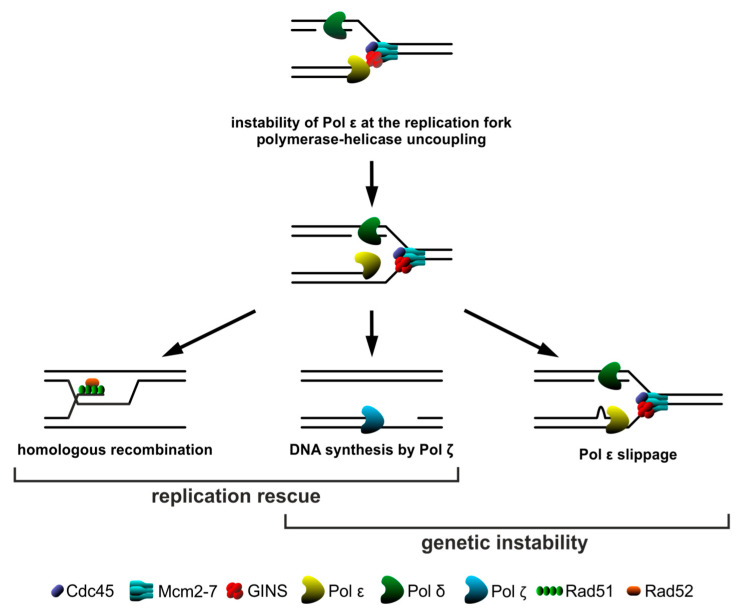
Model showing the consequences of impaired interactions between GINS and Pol ε in the *psf1-100* mutant. Defective interactions within the Cdc45, Mcm2–7, GINS, DNA polymerase epsilon (Pol ε) (CMGE) complex may lead to the instability of leading strand polymerase (Pol ε) at the replication fork and its uncoupling from the helicase. This may result in polymerase slippage or the formation of single-stranded DNA regions. Restart and continuation of DNA synthesis may be ensured by homologous recombination mechanisms; alternatively, ssDNA gaps may be repaired postreplicatively by the error-prone polymerase ζ (Pol ζ).

**Table 1 ijms-21-09484-t001:** Spontaneous mutagenesis rates in *psf1-100* cells with *G2-REV3*.

Genotype	*CAN1* Mutation Rate			*p*-Value ^2^
	× 10^−8^	95% Confidence Intervals	Relative ^1^	
WT	176.0	148.4–247.0	1.0	
*G2*-*REV3*	232.5	216.7–246.1	1.5	0.059951
*rev3Δ*	95.3	88.4–117.6	0.6	0.000855
*psf1-100*	331.9	299.8–415.1	2.0	
*psf1-100 G2*-*REV3*	308.6	301.4–382.3	2.0	0.783131
*psf1-100 rev3Δ*	76.5	69.5–87.8	0.5	0.000017

**^1^** The relative rate is the fold increase in mutability (the rate of mutagenesis in the respective mutant is divided by the rate of mutagenesis in the wild-type). ^**2**^ The Mann–Whitney U test was used for statistical analysis. *p*-values for *G2-REV3* and *rev3Δ* were calculated vs. WT, those for *psf1-100 G2-REV3* and *psf1-100 rev3Δ* were calculated vs. *psf1-100*.

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
