# Peer review of "Recombination and Pol ζ Rescue Defective DNA Replication upon Impaired CMG Helicase—Pol ε Interaction"

_ijms, 2020, doi:10.3390/ijms21249484_

Round 1
Reviewer 1 Report
A nice piece of work is presented in this manuscript. Using a previously constructed mutant psf1-100, in a series of elegant genetic experiments, the authors demonstrated that in Saccharomyces cerevisiae, the impaired interaction between GINS and Pol epsilon requires enhanced involvement Pol zeta and effective recombination as a rescue mechanism for recovery of impaired replication forks. This is an interesting discovery which deserves publication. I have only some minor critical comments:
- L. 102-116: Summary of the obtained results should not be included into Introduction chapter as this disturbs the clarity of presentation. At the end of Introduction, the aim of the study should be described, and rationale of the work can be presented, which is enough to analyse next chapters.
- L.129 and Fig.1a : It should be indicated that the experiments were based on SDS-PAGE and Western-blotting.
- L.130: Insert coma between "used" and "GINS", as otherwise the sentence does not make sense.
- Figs. 2, 3 and 5: Although detailed statistical analyses are provided in aupplementary tables, for clarity of presentation, statisitically significent differences should be indicated by symbols in the figures.
- Fig. 6: Something is wrong with the Y-axis. The scale is interrupted between values 30 and 40, not 40 and 120. Thus, some values can be misinterpreted.
- L. 381: Replace "fork" with "replication fork".
Author Response
Reviewer #1
A nice piece of work is presented in this manuscript. Using a previously constructed mutant psf1-100, in a series of elegant genetic experiments, the authors demonstrated that in Saccharomyces cerevisiae, the impaired interaction between GINS and Pol epsilon requires enhanced involvement Pol zeta and effective recombination as a rescue mechanism for recovery of impaired replication forks. This is an interesting discovery which deserves publication. I have only some minor critical comments:
- 102-116: Summary of the obtained results should not be included into Introduction chapter as this disturbs the clarity of presentation. At the end of Introduction, the aim of the study should be described, and rationale of the work can be presented, which is enough to analyse next chapters.
As suggested by the reviewer, the summary of the research results was removed from the Introduction. Only rationale of the work is presented.
- 129 and Fig.1a : It should be indicated that the experiments were based on SDS-PAGE and Western-blotting.
Lines 134, 135 Fig. 1a has been completed “Western-blot detection of Dpb2 and Psf2 subunits of Pol ε and GINS in protein mixtures separated by SDS-PAGE.”
- 130: Insert coma between "used" and "GINS", as otherwise the sentence does not make sense.
Completed
- 2, 3 and 5: Although detailed statistical analyses are provided in supplementary tables, for clarity of presentation, statistically significant differences should be indicated by symbols in the figures.
In Fig 2, 3 statistically significant differences are indicated by symbols in the figures. The statistically significant differences in the Fig. 5 are explained in the text (lines338-341).
- 6: Something is wrong with the Y-axis. The scale is interrupted between values 30 and 40, not 40 and 120. Thus, some values can be misinterpreted.
The scale on the Fig. 6 has been changed.
- 381: Replace "fork" with "replication fork".
It has been replaced (line 378).

Reviewer 2 Report
The manuscript describes an extensive, multifaceted, and interesting study of the effects of a Psf1 mutation affecting CMG-Pol ε interaction in yeast. The authors combine genetic, biochemical and cell biology approaches to reveal how the cell responds to the defect of leading DNA stand machinery. The allele used, psf1-100, leads to increased spontaneous mutagenesis and recombination. The corresponding protein does not participate in stimulation of replication in vitro as well and the wild-type control. The authors have shown that the replication defect causes accumulation of harmful single-stranded DNA (did they meant ssDNA gaps?), and the trouble is relieved by recombination and participation of DNA polymerase ζ. The experiments are professionally executed, well-presented, and nicely illustrated. The results are of interest to many specialists and would be undoubtedly carefully examined, remembered, and cited. The work is an excellent addition to the literature demonstrating genetic instability and health problems caused by alteration of non-catalytic components of replisome in eukaryotes.
General comments. The manuscript could be more concise. The connection of replication errors and mutations occurring at different cell cycle phases might have been better explained. The striking AT to TA mutator effect of triple psf1-100 pol2-4 rev3 could be better explained. Maybe the context of mutation can give clues?
The authors might consider some editing of the manuscript; we append the list of things that were not clear or surprising to the current reviewer.
Minor comments.
Abstract.
Line 16. Clarify "reciprocal stimulation".
- Edit “employing mutant we demonstrate impaired interaction…”
- Clarify “… enhances incidence of ssDNA”
- What is “level of DNA replication”?
- The evidence for the increase of DNA synthesis by pol ζ is indirect. Maybe the conclusions could be softened.
- Why “extending the leading strand”, beyond telomere?
The introduction could be shortened. Currently, it looks like part of a review paper.
- Conservation is not the only reason.
- The sentence gives the reader a feeling of hopelessness.
- It reads as if there are two different replication machines, but it is just bidirectional replication.
46,48. CMG and GINS abbreviations are explained here, in the abstract, in line 118, and line 693.
- It is not clear "…these genes" in the context of the sentence
- Edit “..both cancer and other genetic diseases”.
- Maybe the phenotype and dumbbell morphology should be more precisely described?
- What kind of ssDNA? Free ssDNA? Gapped DNA?
- There could be a gap in dsDNA. What is the ssDNA gap?
Results, Discussion and Materails.
- Quantification? To what extent weaker?
- The description of the experiment in Fig. 1 is split between the legend and materials, and methods. It is desirable to explain upfront what we see on the gel. Otherwise, the reader has to dig through the Methods section. What is the marker, what was done after incubation of GINS with pol ε, how the signal was detected?
- The gel's image, the basis of Fig. 1b, could be helpful (for example, in Supplementary materials).
- It would be good to see representative images of cells with 0, 1, 2, 3 foci. It looks like decimal points on y-axis in Figs b and c should be periods, not commas.
184 vs. 203. Rad52 – mediator or recombinase?
- typo “fromthe”
- The conclusion sounds soft, "recombination-related" is too broad, and "genome stability or even survival" is undecisive.
- Rephrase “…rate of mutagenesis… was additive”.
- “Pol δ with its non-essential subunit…” sounds uncommon.
231-232. The deletion of the POL32 gene is not benign, so maybe just a combination of two cold-sensitive alleles was too much?
233-234. An impressive result, but for most readers, the rationale of these experiments might be elusive.
- “mutagenic effect of pol ζ on …cells”?
- Interesting section!
244-245. The results are the same as in Jentsch et al. 2010, so maybe Western blots can go to Supplementary files because they confirm that G2-Rev3 works as supposed in the current strain background?
274-276. The results show that Polζ activity in the G2 phase fully supports the mutator effect of psf1-100. The authors should better explain why it “cannot rule out the possibility that Polζ exerts its activity in phase as well”. Otherwise, the last sentence looks isolated.
- Supporting DNA synthesis or supporting replication?
- Do "abnormalities in DNA content" straightforwardly demonstrate "severe genome instability"?
- “Synergistic effect in the mutations rate”?
- Strains or cells accumulate in the S-phase?
- It should not be plural. See: "The noun yeastcan be countable or uncountable. In more general, commonly usedcontexts, the pluralform will also be yeast. However, in more specific contexts, the plural form can also be yeasts, e.g., in reference to various types of yeasts or a collection of yeasts”.
- Why "severely influences"? It is just the elevation of frameshift mutations.
- Not essentially error-free, see:
Elevated mutation rate during meiosis in Saccharomyces cerevisiae.
Rattray et al.PLoS Genet. 2015;11(1):e1004910. PMID: 25569256
Characterizing mutagenic effects of recombination through a sequence-level genetic map. Halldorsson BV, et al., Science. 2019;363(6425):eaau1043.PMID: 30679340
Timing of appearance of new mutations during yeast meiosis and their association with recombination. Mansour O, et al. Curr Genet. 2020 ;66(3):577-592. doi: PMID: 31932974
456-457. Maybe the authors can provide a better explanation of the harmful effect of pol ζ on the viability of psf1-100 strains than "less favorable to cells physiology"?
530-531. The description should be more precise; both cited papers describe a different combination of proteins and substrates.
Author Response
Reviewer #2
The manuscript describes an extensive, multifaceted, and interesting study of the effects of a Psf1 mutation affecting CMG-Pol ε interaction in yeast. The authors combine genetic, biochemical and cell biology approaches to reveal how the cell responds to the defect of leading DNA stand machinery. The allele used, psf1-100, leads to increased spontaneous mutagenesis and recombination. The corresponding protein does not participate in stimulation of replication in vitro as well and the wild-type control. The authors have shown that the replication defect causes accumulation of harmful single-stranded DNA (did they meant ssDNA gaps?), and the trouble is relieved by recombination and participation of DNA polymerase ζ. The experiments are professionally executed, well-presented, and nicely illustrated. The results are of interest to many specialists and would be undoubtedly carefully examined, remembered, and cited. The work is an excellent addition to the literature demonstrating genetic instability and health problems caused by alteration of non-catalytic components of replisome in eukaryotes.
General comments. The manuscript could be more concise. The connection of replication errors and mutations occurring at different cell cycle phases might have been better explained. The striking AT to TA mutator effect of triple psf1-100 pol2-4 rev3 could be better explained. Maybe the context of mutation can give clues?
We have undertaken an attempt to explain the relationship between replication errors and mutations occurring in different phases of the cell cycle in the lines 268-270 and the Discussion section lines 454-462.
We analyzed the sequence context of the observed AT–TA mutations. We do not find any rule in the sequence accompanying the AT–TA changes. Some of them take place in the vicinity of homopolymeric runs, but this is not a rule. Some of the AT – TA mutations are located in the same nucleotides of the CAN1 sequence as in the pol2-4 rev3∆, but with higher frequency. Therefore, it can be that the psf1-100 mutation can affect the fidelity of epsilon polymerase. However, we think that we have too little data to propose an explanation for this phenomenon.
The authors might consider some editing of the manuscript; we append the list of things that were not clear or surprising to the current reviewer.
Minor comments.
Abstract.
Line 16. Clarify "reciprocal stimulation".
Lines 14-16 The sentence has been changed into: “The CMG helicase interaction with the leading strand DNA polymerase epsilon (Pol ε) is essential for the preferential loading of Pol ε onto the leading strand, stimulation of the polymerase, and modulation of helicase activity.”
- Edit “employing mutant we demonstrate impaired interaction…”
Lines 16-18 The sentence has been edited: “Here, we analyze the consequences of impaired interaction between Pol ε and GINS in Saccharomyces cerevisiae cells with the psf1-100 mutation.”
- Clarify “… enhances incidence of ssDNA”
Lines 20-21 The sentence has been clarified: “It also increases the occurrence of single-stranded DNA (ssDNA) gaps and the demand for homologous recombination.”
- What is “level of DNA replication”?
Lines 18-20 The sentence has been changed into: “This significantly affects DNA replication activity measured in vitro, while in vivo, the psf1-100 mutation reduces replication fidelity by increasing slippage of Pol ε, which manifests as an elevated number of frameshifts.”
- The evidence for the increase of DNA synthesis by pol ζ is indirect. Maybe the conclusions could be softened.
Lines 22-23 The sentence has been changed into: “Additionally, we observe increased participation of DNA polymerase zeta (Pol ζ) in DNA synthesis.”
- Why “extending the leading strand”, beyond telomere?
Lines 57-59 The sentence has been modified: “Pol ε is responsible for replication of the leading strand, but was additionally postulated to link DNA replication with the S-phase checkpoint in S. cerevisiae.”
The introduction could be shortened. Currently, it looks like part of a review paper.
The Introduction has been shortened, it is 1029 words long instead of 1214
- Conservation is not the only reason.
This sentence has been removed
- The sentence gives the reader a feeling of hopelessness.
This sentence has been removed
- It reads as if there are two different replication machines, but it is just bidirectional replication.
Lines 45-47 The sentence has been modified: “Replication machineries assembled at the replication origin move by the action of 11-subunit CMG helicase (Cdc45-Mcm2-7-GINS), which consists of the heterohexameric Mcm2-7 helicase motor ring, Cdc45 and the GINS heterotetramer (Psf1, Psf2, Psf3, Sld5).”
46,48. CMG and GINS abbreviations are explained here, in the abstract, in line 118, and line 693.
We deleted the explanation from line 118 (now 106) and decided that a repeated explanation of the abbreviations in the Introduction (lines 45-47) will enable a better understanding of the research subjects for those who are not familiar with nomenclature of replisome components. We also left the explanation in the abbreviations list required by the journal.
- It is not clear "…these genes" in the context of the sentence
Lines 89-91 The sentence has been clarified: “The study of noncatalytic proteins of the CMGE complex is attracting greater attention because mutations in genes encoding its subunits or changes in their expression have a significant impact on the development of cancer or genetic diseases.”
- Edit “..both cancer and other genetic diseases”.
Lines 89-91 The sentence has been edited: “The study of noncatalytic proteins of the CMGE complex is attracting greater attention because mutations in genes encoding its subunits or changes in their expression have a significant impact on the development of cancer or genetic diseases.”
- Maybe the phenotype and dumbbell morphology should be more precisely described?
Lines 98-100 The sentence has been changed into: “The psf1-100 strain exhibits a cold-sensitive phenotype (inability to grow at 18˚C) and dumbbell cell morphology with the nucleus in the isthmus between mother and daughter cell.”
- What kind of ssDNA? Free ssDNA? Gapped DNA?
As a result of introduction shortening this part has been removed.
- There could be a gap in dsDNA. What is the ssDNA gap?
As a result of introduction shortening this part has been removed.
Results, Discussion and Materails.
- Quantification? To what extent weaker?
Lines 116-121 In agreement with the Reviewers suggestion, we modified the description of the results: “In the control reaction, we detected bands representing the subunits of both GINS and Pol ε (Figure 1a), demonstrating that under the experimental conditions that were used, GINS binds Pol ε. Similar results were observed for the Psf1-1 mutant subunit (R84G) - the band intensity was 0.96 compared to Psf1. In contrast, when GINS contained the Psf1-100 subunit, the band intensity was 0.31, compared to Psf1, while in absence of Psf1 the band intensity was 0.11 (negative control). This demonstrates impaired interaction between GINSPsf1-100 and Pol ε.”
- The description of the experiment in Fig. 1 is split between the legend and materials, and methods. It is desirable to explain upfront what we see on the gel. Otherwise, the reader has to dig through the Methods section. What is the marker, what was done after incubation of GINS with pol ε, how the signal was detected?
We have added necessary information to the main text and figure legend:
Lines 114-116 “After incubation, protein mixtures were separated on SDS-PAGE, followed by immunodetection of Dpb2 and Psf2 subunits.”,
Lines 132-136 “(A) Western-blot detection of Dpb2 and Psf2 subunits of Pol ε and GINS in protein mixtures separated by SDS-PAGE. Two pmol of various GINS complexes (WT, Psf1-1 and Psf1-100) containing Halotag-Sld5 were fixed on 10 µl of Halo Link Resin and then incubated with 2 pmol of Pol ε in 400 µl buffer at 30°C for 30 min. Input of GINS and Pol ε were 0.5 and 0.05 pmol, respectively. Precision Plus Protein™ All Blue Prestained Protein Standard was used as molecular weight marker.”
Lines 527-529 “Separated proteins were transferred to membranes and incubated with anti-Psf2 [48] and anti-Dpb2 [128] antibodies detected using an Odyssey infrared imaging system (LI-COR).”.
The gel's image, the basis of Fig. 1b, could be helpful (for example, in Supplementary materials).
Lines 132-141 The description of the experiments in Fig,1 has been modified. The gel image has been added to the Supplementary materials Fig. A2.
- It would be good to see representative images of cells with 0, 1, 2, 3 foci. It looks like decimal points on y-axis in Figs b and c should be periods, not commas.
According to the Reviewer suggestion representative images of cells with Rfa1 foci have been added to the Supplementary materials Fig. A3. The commas in the decimal points on y-axis in Figures b and c has been changed to periods.
184 vs. 203. Rad52 – mediator or recombinase?
Lines 173-174 The sentence has been changed: “Rad52, as a recombination mediator, promotes HR by facilitating the exchange of ssDNA-bound RPA with the Rad51 protein.
Lines 191-192 “Next, we analyzed the genetic interaction between the psf1-100 mutation and deletion of the RAD52 gene encoding the mediator of recombination.”
- typo “fromthe”
has been corrected (line 185)
- The conclusion sounds soft, "recombination-related" is too broad, and "genome stability or even survival" is undecisive.
Lines 199-200 The sentence has been modified: “Together, these results points-out the involvement of recombination processes in the maintenance of genomic stability in psf1-100 mutant cells.”
- Rephrase “…rate of mutagenesis… was additive”.
Lines 208-210 The sentence has been rephrased: “The rate of CanR mutagenesis in the double psf1-100 mms2Δ showed an additive effect (388x10-8) compared to the single mutants: 182x10-8 for psf1-100 and 282x10-8 for mms2Δ (Figure 2g).”
- “Pol δ with its non-essential subunit…” sounds uncommon.
Line 213-214 the phrase has been modified “This process depends on the activity of the Pif1 helicase and Pol δ “ and the requirement of Pol32 for BIR is now mentioned line 219 “…POL32 encoding a subunit of Pol δ required for BIR.”
231-232. The deletion of the POL32 gene is not benign, so maybe just a combination of two cold-sensitive alleles was too much? 233-234. An impressive result, but for most readers, the rationale of these experiments might be elusive.
Lines 221-228 In agreement with the Reviewer suggestion, we modified the rationale to make it clearer. Now it says: “To exclude the possibility that this lethal effect results from increased participation of Pol δ in DNA replication in psf1-100 cells, we used the pol3-5DV variant of Pol δ which is impaired in proof-reading activity [94]. We analyzed the mutagenesis rates in the double mutant psf1-100 pol3-5DV in both MMR-deficient or MMR-proficient background. (Figure A5). The non-synergistic effect in MMR-deficient background may shows that the participation of Pol δ in DNA replication in psf1-100 cells is not increased. Similarly, the same effect in MMR-proficient background suggests that the involvement of Pol δ in repair synthesis is neither enhanced at least at the level detectable in this genetic test.”
- “mutagenic effect of pol ζ on …cells”?
Lines 230-2331 Has been corrected: “The mutagenic effect of Pol ζ may be exerted through its activity during DNA replication in the S phase or repair mechanism, which operates mainly in the G2 phase.”
- Interesting section!
244-245. The results are the same as in Jentsch et al. 2010, so maybe Western blots can go to Supplementary files because they confirm that G2-Rev3 works as supposed in the current strain background?
Since we have introduced the Clb2-Rev3 fusion into different strain background we left the Western blots in the Results section.
274-276. The results show that Polζ activity in the G2 phase fully supports the mutator effect of psf1-100. The authors should better explain why it “cannot rule out the possibility that Polζ exerts its activity in phase as well”. Otherwise, the last sentence looks isolated.
Because the statement “it cannot rule out the possibility that Pol ζ exerts its activity in phase S as well” was unclear, we removed it (line 268).
- Supporting DNA synthesis or supporting replication?
Lines 276-277 The sentence has been corrected: “Our results presented above demonstrate that both Pol ζ-dependent DRIM and Rad51-dependent DNA repair pathways are involved in supporting DNA replication in psf1-100 cells.”
- Do "abnormalities in DNA content" straightforwardly demonstrate "severe genome instability"?
Lines 296-298 The sentence has been modified: “In contrast, deletion of both REV3 and RAD51 in psf1-100 cells resulted in abnormalities in the DNA content, suggesting genomic instability (Figure 4b).”
- “Synergistic effect in the mutations rate”?
Lines 323-327 We apologize for this confusing statement. The sentence has been corrected: “The results of mutation rate analyzes obtained for psf1-100 pol2-4 rev3Δ were compared with spectra of rev3Δ, psf1-100 rev3Δ, and pol2-4 rev3Δ (Figure 5 and Table A7). We observed a synergistic effect of pol2-4 and psf1-100 alleles in the rev3Δ background (Table A7), suggesting that two different mechanisms act in concert to influence the pool of errors.”
- Strains or cells accumulate in the S-phase?
Line 387 The sentence has been corrected: “Cells with the psf1-100 allele accumulate in S phase (Figure 4b).”
- It should not be plural. See: "The noun yeast can be countable or uncountable. In more general, commonly used contexts, the plural form will also be yeast. However, in more specific contexts, the plural form can also be yeasts, e.g., in reference to various types of yeasts or a collection of yeasts”.
The sentence (line 351) has been corrected as suggested.
- Why "severely influences"? It is just the elevation of frameshift mutations.
Lines 405-407 The sentence has been modified: “Therefore, it cannot be excluded that impaired interaction between Dpb2 of Pol ε and Psf1-100 of GINS affects the directing of the leading strand from the helicase to the polymerase.”
- Not essentially error-free, see:
Elevated mutation rate during meiosis in Saccharomyces cerevisiae.
Rattray et al.PLoS Genet. 2015;11(1):e1004910. PMID: 25569256
Characterizing mutagenic effects of recombination through a sequence-level genetic map. Halldorsson BV, et al., Science. 2019;363(6425):eaau1043.PMID: 30679340
Timing of appearance of new mutations during yeast meiosis and their association with recombination. Mansour O, et al. Curr Genet. 2020 ;66(3):577-592. doi: PMID: 31932974
Lines 420-421 We agree that the recombination may be error-prone, therefore the statement that “it is generally error-free” has been removed from the manuscript. The sentence has been changed: “Recombination is frequently coupled with a progressing replication fork allowing its repairs and restart. Yeast Rad51 and Rad52 recombinases were detected in both unperturbed and stressed forks.”
456-457. Maybe the authors can provide a better explanation of the harmful effect of pol ζ on the viability of psf1-100 strains than "less favorable to cells physiology"?
Lines 443-451 We have changed the text to make it clear: “Although both recombination and Pol ζ allow continuation of DNA synthesis in the psf1-100 strain, inactivation of the later, increases the survival of the psf1-100 strain by approximately 40%, as observed for CFU counts in Figure 4a, and reduces the observed level of mutagenesis to the level of the rev3Δ strain (Table 1). To explain these observations we can speculate, that besides the error-prone nature of Pol ζ, DNA repair processes involving this polymerase, due to its lower processivity can be less extensive than those based on recombination. Interestingly, deletion of REV3 in the psf1-100 rad51Δ strain causes a drastic decrease in CFU counts (Figure 4a). This demonstrates that both switching to Pol ζ and recombination are mechanisms that are employed to fulfill genetic material duplication in psf1-100 cells.”
530-531. The description should be more precise; both cited papers describe a different combination of proteins and substrates.
Lines 531-548 A detailed description of the assay has been added:
4.5. In vitro replication assay
pARS1 (245 bp ARS1 fragment cloned into the SmaI site of pNEB193) DNA was mixed with the loading complex composed of 11 nM ORC, 23 nM Cdc6, and 50 nM MCM–Cdt1 in a buffer containing 25 mM HEPES-KOH at pH 7.6, 100 mM Kglutamate, 10 mM magnesium acetate, 0.01% NP-40, 100 μg/ml of BSA, 1 mM DTT, and 5 mM ATP. After 20 min of incubation at 30°C, DDK was added directly to the reaction to a final concentration of 26 nM and incubation was continued at 30°C for 20 min. Next, the replication complex proteins and solutions: were added directly to the reaction: 25 nM Sld3–Sld7, 30 nM Sld2, 30 nM Dpb11, 30 nM GINS, 40 nM Cdc45, 20 nM Polε, 5 nM Mcm10, 100 nM RPA, 3.4 nM CDK, 5 nM Polα, 20 nM RFC, 20 nM PCNA, 10 nM Top2, 10 nM Polδ, 20 mM each NTP, 8 mM dATP, dCTP, and dGTP, 6 mM dTTP, and 1 mM Biotin-16-dUTP (Sigma-Aldrich). After incubation at 30°C for 20 min, the reaction was terminated by the addition of 1/5 volume of Alkaline stop Dye (Loading Buffer) containing 0.3 N NaOH, 6 mM EDTA, 36% glycerol, and 0.1% Orange G dye. The products were separated on 1% alkaline agarose gels in 0.05 N NaOH and 1 mM EDTA for 75 min at 75 V. DNA was transferred from the gels to Hybond N+ membranes (GE) in 0.5 TBE Buffet for 30 min at 80 V. The membrane was treated with 20 x SCC and crosslinked in UV linker. After treatment with 2% ECL Advance Blocking Reagent in TBST for 10 min and two 5 min washes by TBST, the membranes was incubated for 30 min with 0.5 μg/ml of IR-Dye 680 RD streptavidin (LICOR) in TBST with 10% SDS. After washing in TBST with 0.1% SDS, the membrane was scanned on an Odyssey infrared imaging system (LI-COR).
